# *Candida albicans* Shields the Periodontal Killer *Porphyromonas gingivalis* from Recognition by the Host Immune System and Supports the Bacterial Infection of Gingival Tissue

**DOI:** 10.3390/ijms21061984

**Published:** 2020-03-14

**Authors:** Dominika Bartnicka, Miriam Gonzalez-Gonzalez, Joanna Sykut, Joanna Koziel, Izabela Ciaston, Karina Adamowicz, Grazyna Bras, Marcin Zawrotniak, Justyna Karkowska-Kuleta, Dorota Satala, Andrzej Kozik, Edyta Zyla, Katarzyna Gawron, Katarzyna Lazarz-Bartyzel, Maria Chomyszyn-Gajewska, Maria Rapala-Kozik

**Affiliations:** 1Department of Comparative Biochemistry and Bioanalytics, Faculty of Biochemistry, Biophysics and Biotechnology, Jagiellonian University in Krakow, 30-387 Krakow, Poland; dominika.ostrowska7@gmail.com (D.B.); miriam.gonzalez.gonzalez93@gmail.com (M.G.-G.); asia.sykut@gmail.com (J.S.); grazyna.cholewa@uj.edu.pl (G.B.); marcin.zawrotniak@uj.edu.pl (M.Z.); justyna.karkowska@uj.edu.pl (J.K.-K.); 2Department of Microbiology, Faculty of Biochemistry, Biophysics and Biotechnology, Jagiellonian University in Krakow, 30-387 Krakow, Poland; joanna.koziel@uj.edu.pl (J.K.); izabela.ciaston@uj.edu.pl (I.C.); karina.adamowicz@uj.edu.pl (K.A.); 3Department of Analytical Biochemistry, Faculty of Biochemistry, Biophysics and Biotechnology, Jagiellonian University in Krakow, 30-387 Krakow, Poland; dorota.satala@uj.edu.pl (D.S.); andrzej.kozik@uj.edu.pl (A.K.); 4Department of Cell Biochemistry, Faculty of Biochemistry, Biophysics and Biotechnology, Jagiellonian University in Krakow, 30-387 Krakow, Poland; zyla.edyta@doctoral.uj.edu.pl; 5Department of Molecular Biology and Genetics, School of Medicine in Katowice, Medical University of Silesia, Medykow 18, 40-752 Katowice, Poland; katgaw1@gmail.com; 6Department of Periodontology and Oral Medicine, Faculty of Medicine, Jagiellonian University, Medical College, 31-155 Krakow, Poland; katarzyna.lazarz-bartyzel@uj.edu.pl (K.L.-B.); mdgajews@cyf-kr.edu.pl (M.C.-G.)

**Keywords:** *Candida albicans*, *Porphyromonas gingivalis*, biofilm, infection, periodontitis, macrophages, fibroblasts

## Abstract

*Candida albicans* is a pathogenic fungus capable of switching its morphology between yeast-like cells and filamentous hyphae and can associate with bacteria to form mixed biofilms resistant to antibiotics. In these structures, the fungal milieu can play a protective function for bacteria as has recently been reported for *C. albicans* and a periodontal pathogen—*Porphyromonas gingivalis*. Our current study aimed to determine how this type of mutual microbe protection within the mixed biofilm affects the contacting host cells. To analyze *C. albicans* and *P. gingivalis* persistence and host infection, several models for host–biofilm interactions were developed, including microbial exposure to a representative monocyte cell line (THP1) and gingival fibroblasts isolated from periodontitis patients. For in vivo experiments, a mouse subcutaneous chamber model was utilized. The persistence of *P. gingivalis* cells was observed within mixed biofilm with *C. albicans*. This microbial co-existence influenced host immunity by attenuating macrophage and fibroblast responses. Cytokine and chemokine production decreased compared to pure bacterial infection. The fibroblasts isolated from patients with severe periodontitis were less susceptible to fungal colonization, indicating a modulation of the host environment by the dominating bacterial infection. The results obtained for the mouse model in which a sequential infection was initiated by the fungus showed that this host colonization induced a milder inflammation, leading to a significant reduction in mouse mortality. Moreover, high bacterial counts in animal organisms were noted on a longer time scale in the presence of *C. albicans*, suggesting the chronic nature of the dual-species infection.

## 1. Introduction

The unique habitat within the oral cavity enables the co-existence of various microbial species, providing them with an abundance of nutrients, variable oxygen availability and diverse surfaces (e.g., gums and teeth) for convenient colonization [1]. The matured oral microbiome consists of hundreds of microbial species that form a complex community known as biofilm [2]. Biofilm constituents include bacteria, fungi, viruses, archaea and protozoa [1,3]. In healthy individuals, the oral cavity is colonized mostly by Gram-positive facultative anaerobic bacteria that exist in a proper equilibrium with the host immune responses [4,5]. However, a microbiome imbalance resulting from improper oral hygiene, with the continuous formation of dental plaques in gingival tissues, contributes to oral diseases ranging from reversible gingivitis to irreversible periodontitis and results in the destruction of periodontal ligaments and alveolar bone [6]. Moreover, periodontitis may enhance the risk for developing systemic inflammatory diseases like atherosclerosis, infective endocarditis, diabetes, adverse pregnancy outcomes, respiratory diseases, and rheumatoid arthritis [7,8,9,10,11]. The keystone pathogen of periodontitis is *Porphyromonas gingivalis* which collaborates with the other anaerobic species, such as *Tannerella forsythia* and *Treponema denticola*, belonging to the traditional “red complex”, subsequently forming a pathogenic dental pocket that results from the progressive degradation of periodontal tissues [2]. However, periodontitis progression depends also on individual risk factors such as age, lifestyle, dental hygiene, smoking, and some systemic disorders [12,13,14]. Nevertheless, the main determinants of this disease’s onset are the quality and quantity of the microorganisms involved in this process which can exert mutually antagonistic or cooperative interactions, attributable to metabolite gradients and quorum sensing [15,16,17,18].

Most prior reports on periodontal diseases have limited their focus to the oral bacteriome and few have indicated a possible role of the fungal microbiome. This limitation is partially due to the difficulties in DNA extraction, PCR amplification, and inconsistent fungal nomenclature. In the oral biofilm, fungi can occur as transient members [19] or as definite constituents [20], depending on the condition of the host (see the review in Reference [21]). Moreover, a positive correlation between fungal and bacterial species in the oral cavity was demonstrated previously by Persoon et al. [22]. Subgingival fungi are present in 10–30% of healthy hosts [20,23] and in up to 52% of diabetics [20,24,25]. However, a prior comparison of the mycobiome between healthy persons and patients suffering from periodontal disease indicated no significant differences in terms of fungal abundance [26].

One of the most common fungal species in the oral cavity and which is also prevalent in the periodontal pockets of periodontitis patients is *Candida albicans* [25,27]. This yeast-like fungus uses morphological and physiological changes to adapt to variable conditions in its inhabited niches in either healthy or immunocompromised human hosts. These adaptive responses enhance its survival and enable changes from a colonizer to a pathogen. *Candida albicans* cells can exist in yeast-like or filamentous (hyphal) forms, both of which are involved in host colonization. The yeast form is considered to be important for disseminated blood infections, and the hyphae contribute to the invasion of host cells [28] and are a critical feature of pathogenicity at the mucosal surface [29].

Both morphological forms of *C. albicans* are equipped with multiple virulence factors, including adhesins and invasins located on the cell surface and extracellular hydrolytic enzymes capable of degrading host proteins and lipids [30]. The types and amounts of virulence factors change with the morphology, phenotyping, and the progress of the infection, enabling efficient host colonization and high resistance to antifungal agents [31].

Studies concerning the possible interaction of obligate anaerobes with *C. albicans* in the oral cavity were prompted by the seminal finding that fungi could deplete the oxygen level within the biofilm formed by mixed-species microorganisms [32]. Several recent reports have identified possible interactions between *P. gingivalis* and *C. albicans* that can allow the cooperation of both microorganisms for mutual biofilm development and host invasion. For example, it was observed that *P. gingivalis* influences *C. albicans* morphology, enhancing germ tube formation [33]. These findings were supported by the observed increased expression of genes encoding the *C. albicans* main adhesins, Als3 and Hwp1, and a secreted aspartic protease 6 (Sap6) that correlated with hyphal morphology [34]. However, some opposing effects have also been observed [35,36]. The mutual contact of both microorganisms was found to be based on direct interactions between the fungal adhesin Als3 and the adhesive domain of gingipain RgpA [34], as well as the virulence factor ItlJ belonging to the internalin protein family of *P. gingivalis* [37]. Another conductive interaction was also determined for the adhesion among both pathogens that induced the type 9 secretion system of *P. gingivalis* and increased the pathogenicity of the community [38].

On the other hand, the importance of a bacterial extracellular enzyme peptidylarginine deiminase (PPAD) for the mutual contact of both pathogens has been proposed [39]. This enzyme converts protein arginine residues to citrullines, and this modification of selected surface-exposed *C. albicans* proteins was identified during the formation of mixed biofilms by both microorganisms under hypoxic and normoxic conditions. Quantitative estimations of the bacterial adhesion to fungal cells demonstrated the importance of PPAD activity in this process, since the level of binding of a *P. gingivalis* mutant strain deprived of PPAD was significantly lower than that observed for the wild-type strain. Moreover, attenuated macrophage responses (i.e., a decreased production of selected cytokines and chemokines) were detected upon mixed infection with *P. gingivalis* and *C. albicans*. A similar effect was also observed for the mixed biofilm formed between *F. nucleatum* and *C. albicans*, where the co-aggregation of both microorganisms reduced the production of MCP-1 and TNF-α. Such mutualistic protection from microbe killing by phagocytes was proposed as the mechanism underlying increased bacterial persistence [40].

The aim of our current study was the in vitro and in vivo identification of the possible preservation of *P. gingivalis* by *C. albicans* cells. This was postulated to occur either through just contact with the fungi or due to the formation of a mixed-species biofilm that protects invading microbes from host recognition and/or facilitates further chronic host infection.

## 2. Results

Microbial multispecies biofilms developing in the periodontium encounter various host cell types including epithelial and inflammatory cells. *P. gingivalis* initiates the infection of a host through its adherence to and internalization by epithelial cells [41]. On the other hand, macrophages that control the bacterial or fungal burden during early infection are particularly critical to a host’s ability to counteract microbial infection [42,43]. In our present study, we investigated the response of macrophages, represented by phorbol 12-myristate 13-acetate (PMA) stimulated THP-1 cells, and fibroblasts acquired from healthy persons and periodontitis patients to contact with mixed bacterial–fungal biofilms.

### 2.1. Collaboration of P. gingivalis Cells with a Fungal Partner during Infection Weakens the Alertness of the Host Defense Response by Macrophages

The monocytes and neutrophils found in abundance in the gingival tissue during periodontal disease respond to microbial stimuli, among others, via cytokine and chemokine production [44]. In our previous report [45], we used THP-1 cells to demonstrate the influence of mixed biofilm formation on host cell genetic responses. We found that the expression of genes encoding TNFα, IL-1β, IL-6, IL-10, and monocyte chemoattractant protein-1 (MCP-1) was subject to various changes during the THP-1 cell contact with a mixed biofilm, formed between *C. albicans* and *P. gingivalis* cells, in comparison to the responses of host cells contacting only bacteria [39]. However, the mechanism underlying such changes has remained unclear. They can result from some antagonistic interactions between pathogens or from the bacterial cells protection by fungal biofilm. The primary triggers of host cytokine responses to *P. gingivalis* include LPS and major proteolytic enzymes of the bacterium, the gingipains. However, gingipains are also known to function in the proteolytic degradation of cytokines and chemokines thus preventing the resolution of infection [46,47]. To verify the possible role of gingipains in the responses of THP-1 cells to contact with a dual-species biofilm, in our current studies, we analyzed specific cytokine and chemokine production by THP-1 cells, treated with the supernatants acquired after centrifugation of mono- or dual-species biofilms, formed by fungal and bacterial cells. This approach should simplify the interaction model (Figure 1).

The changes in the protein levels of the Il-1β, TNF-α, and Il-8 were determined by ELISA, testing the samples obtained from THP-1 cells that contacted supernatants for 3 and 24 h at 37 °C. The analysis revealed the significant differences in Il-1β production after 24 h of host cell interaction with the secreted components of biofilm. The response of THP-1 cells to only bacterial biofilm supernatants resulted in a 4-fold lower Il-1β production, compared to the responses of THP-1 to contact with fungal biofilm supernatant (100%). However, THP-1 interaction with mixed microbial biofilm caused a 25-fold increase in IL-1β abundance.

The activation of IL-8 production by THP-1 cells following contact with supernatants was identified at different levels, depending on the type of mono-species microbial–host interaction. The significant disappearance of IL-8 production was observed, however, in response to the treatment of host cells for 24 h with the mixed fungal–bacterial biofilm components.

In contrast, the production of TNF-α in the response of host cells to 3 h of contact with supernatant from mixed biofilm increased 1.5-fold in comparison to the THP-1 cell responses to mono-species bacterial biofilm, with little influence of *C. albicans* on the generation of this cytokine. However, during longer interaction of the THP-1 cells with microbial supernatants (24 h), the TNF-α level decreased dramatically in comparison to the responses to mono-species bacterial biofilm constituents secreted to the medium.

An explanation for why different changes in cytokine production by THP-1 cells occured in contact with soluble components secreted by formed polymicrobial biofilm requires a consideration of the proteolytic properties of the main virulence factors of bacteria, the gingipains, towards these host defense compounds. Moreover, the proteolytic properties of bacterial cells can be modulated during mutual contact with *C. albicans*. To verify these possibilities, the activity of the arginine-specific gingipain (Rgp) was tested in supernatants obtained from mono- and dual-species biofilms (Figure 2A) formed under different levels of oxygen availability, as appropriate to the microbes in question. Indeed, the activity of Rgp increased significantly in the presence of *C. albicans* cells up to 130% under anoxic conditions, and even to 10-fold under the normoxic conditions, preferred by *C. albicans* for growth, but changing within the fungal biofilm milieu as was mentioned by Fox et al. (2014) [32]. The increased production of Rgp was also confirmed by Western blot analysis (Figure 2C,D). Our present data suggest that a fungal biofilm generates protective conditions for bacterial cells and can influence not only their recognition by macrophages but enable them to induce the protease virulence factors that can effectively destroy cytokine molecules, thereby weakening the host alertness system. However, the issue of whether the augmentation of protease activity results from an increasing number of viable bacterial cells, protected by fungal biofilm, or, as we also demonstrated in Figure 2B, from the increasing expression of the gene that encodes this enzyme, remains to be resolved.

### 2.2. A Mouse Model of P. gingivalis Host Sensing during Mutual Infection with C. albicans Confirms the Bacterial Protection from Host Recognition by These Fungal Cells

Considering the observed responses of selected host cells to dual-species infection or secondary infection involving *P. gingivalis* or *C. albicans*, we noted variations in these responses that depended on the condition of the host cells and the infecting pathogens. To mimic the dynamics of the infection process, we used the well-established murine subcutaneous chamber model, which has been utilized in previous studies focused on the virulence of periodontal species [48,49].

We analyzed two co-infection scenarios where the pathogens invaded the host at the same time or when the well-settled fungi provided an environment for *P. gingivalis* colonization. As the data obtained using both models were comparable, we here present only the results of the sequential infection process.

The analysis of animal survival showed that while the infection with *P. gingivalis* was lethal for 21% of the tested animals, pre-infection with *C. albicans* reduced this mortality rate to 7% (Figure 3A). Mono-infection with *C. albicans* caused no animal deaths.

We then examined the bacterial and fungal load in the chamber fluid by plating and colony counting. The number of recovered colonies showed significantly impaired bacterial cell quantity in the *C. albicans* companion compared to the bacterial mono-infection measured within two days post-infection (Figure 3B).

It was notable that the number of bacteria was maintained at a high level in the mono- and dual-species infection models up to day 4. In the case of mono-infection starting from the third day, we observed either animal deaths (Figure 3A) or a gradual eradication of the pathogen from the chamber (Figure 3B,C). The bacterial incidence in the chamber fluid decreased during this time to 24% of surviving animals infected with bacteria, whilst in the dual-species infection model, the incidence of bacteria in the chamber remained at 85% (Figure 3C). Taken together, our data indicated for the first time that *P. gingivalis* was protected by *C. albicans* in the host environment. Notably, the *C. albicans* infection also changed during mutual contact of these microbes with the host. In this case, the fungal cell presence in the chamber fluid decreased during mono-infection, but the interaction with bacteria enhanced the fungal cell proliferation during the next 2–3 days with a slow decrease occurring on the fourth day (Figure 3D). The stimulation of fungal cell growth was also previously noted during dual-species biofilm formation on an artificial surface [34], but the “stimulator” remains unknown.

We already documented, using in vitro analysis, that *C. albicans* significantly reduced the response of leukocytes to *P. gingivalis* (Figure 1). We examined this phenomenon also in vivo by analyzing the inflammatory response of leukocytes in the chamber fluid. The activity of the neutrophil granular enzymes: elastase (NE) and myeloperoxidase (MPO) showed a significant increase in this fluid, especially NE during the second and third day of bacterial infection (Figure 4). A 4-fold increase of NE activity during this period was observed compared to the first day, whereas the activity of NE in the chamber fluid from the dual-species infection remained more or less at the same level as the first day of infection, and was 2-fold lower than the responses detected for the mono-species, bacterial contact during 24 h. In addition, the changes in MPO activity showed a similar decreasing tendency, although with lower significance, when the comparison was made between the dual-species and mono-species bacterial infection. The results once again revealed the impairment of *P. gingivalis* recognition by the host when these bacteria formed the mixed biofilm with *C. albicans*.

The local inflammatory reactions induced by pathogens are often manifested by systemic reactions. We therefore examined our experimental mice for changes in body weight, blood morphology, and circulated inflammatory mediators. As we observed some symptoms of leukopenia, we expanded our analysis to the systemic dissemination of the tested microorganisms. The results revealed the distribution of both *P. gingivalis* and *C. albicans* to the spleen and kidney of the infected animals. Detailed analysis revealed an elevated bacterial load (Figure 5A,C) and a higher incidence of infection (Figure 5B,D) in organs isolated from mice infected solely with *P. gingivalis* compared to the animals exposed to mixed microbial species. These data were consistent with the observed mortality rates shown in Figure 3. Our results thus indicated that *C. albicans* can promote the local infection of *P. gingivalis* and thereby attenuate its systemic distribution.

### 2.3. Altered Susceptibility of Periodontal Patient Fibroblasts to Fungal Infection Compared to Those from Healthy Donors

To date, we considered the consequences of prior colonization of host cells or tissues by *C. albicans* and its implications for *P. gingivalis* infection. However, there is no information as to whether prolonged bacterial infection of host cells during periodontitis may promote secondary fungal invasion. For this purpose, we evaluated the susceptibility and responses of fibroblasts isolated from healthy donors (HD, control group A) and from the tissues of periodontitis patients (PP, group B) to *C. albicans* colonization. Fibroblasts are the predominant cell type in periodontal tissue and play important roles not only in tissue regeneration but also in the inflammatory response associated with pathogenic cell invasion.

To evaluate the consequences of prolonged contact between fibroblasts and periodontal pathogens on the efficiency and sensitivity of both HD and PP fibroblast types to fungal colonization, we tested their proinflammatory cytokine production. For this purpose, the main *C. albicans* virulence factors were used as these factors can trigger host cell responses, including mannans or surface located mannoproteins (CWP), glucans, and either secreted Saps (Sap3, Sap6) or those bound to the fungal cell surface (Sap9). We used these isolated compounds for the treatment of fibroblasts instead of whole fungal cell contact to avoid the impacts of the attenuated adhesion of fungi to periodontitis fibroblasts described above. The most intensive production of signaling molecules by both types of fibroblasts was noted for the response to protease treatment lasted 6 h (Figure 6). However, the PP fibroblasts showed a 4-fold lower level of IL8 and 3-fold lower IL-6 production compared to the HD fibroblasts. Lower IL1β and TNFα production in the PP cells was also detected. The responses to contact with mannans, glucans, or mannoproteins did not differ significantly between the HD and PP fibroblasts. Under all experimental conditions, the monolayer formation and viability of both fibroblast types were tested but no significant differences were found.

## 3. Discussion

The morphologic and phenotypic diversity of *C. albicans* cells enables them to form a biofilm that provides a high level of resistance to the host defense response, antifungal drugs, or microenvironmental changes. This excellent adaptation of *C. albicans* to the prevailing conditions enables other microorganisms, especially bacteria, to benefit by interacting with fungi to increase their own chances of survival in an unfavorable environment or to enhance their invasiveness [50].

One of the seminal observations of this type of protective interaction is between *P. gingivalis*, the well-known anaerobic periodontal pathogen, and the fungus *C. albicans*, likely due to the hypoxic microenvironment that is provided by a mixed-species biofilm formation [34]. The model arising from the study of *Bacteroides fragilis* and *Clostridium perfringens* of a “mini-biofilm” formation [32] may also be relevant to our current observations, where free-floating cellular aggregates resembling miniature biofilms enable *P. gingivalis* to proliferate under otherwise toxic conditions. The mutual contact of *P. gingivalis* and *C. albicans* that resulted in increased viability of bacteria was considered in detail in our previous reports [34,39]. In our present analyses, we demonstrate the biological consequences of mixed biofilm formation for the host cells and their responses.

The development of a mixed biofilm with *P. gingivalis* and *C. albicans* can naturally take place in various combinations, depending on the progress of the infection, and both pathogens could conceivably settle the new host niche and invade host cells. Periodontal disease could develop in a patient suffering from candidiasis or a patient with periodontitis could suddenly develop a fungal infection. In these instances, the host cells are already pre-stimulated with the infectious process and respond to the secondary infection, often with consequences arising from mutual pathogen interactions.

The first of these scenarios was considered in our analysis of the responses of THP-1 cells to a mixed infection, where supernatant-challenged macrophage-like THP-1 cells, responded with different cytokine production levels, compared to the mono-species infection. First, we considered the primary THP-1 reaction, represented by the secretion of the IL-1β cytokine that appears shortly after the microbial challenge and mediates the release of “secondary” proinflammatory cytokines such as IL6 and IL8 in an autocrine manner [46,47].

Previous clinical studies on gingival crevicular fluid (GCF) have shown that proinflammatory cytokines are elevated at sites exhibiting clinical signs of inflammation [51]. IL-1β is found at higher concentrations in the GCF of patients with periodontitis compared to healthy controls [52,53]. In contrast, IL-8 is at lower concentrations in the GCF of patients with aggressive periodontitis [54]. In our current experiments, both mono-species microbial contacts with THP-1 cells resulted in an increased production of IL-1β but to a lesser extent with the bacteria. Although IL-1β is more resistant to gingipains compared to other cytokines, it is still the substrate for Lys-specific gingipain [47]. The biofilm or any aggregate formed between *C. albicans* and *P. gingivalis* cells, while interacting with THP-1 triggered a significant increase in Il-1β production compared to mono-species infection, suggesting the cooperation of both microbes towards the host cells.

The secretion of IL-1β by monocytes and macrophages can engage multiple mechanisms that may have specific contributions to IL-1β release depending on the type or strength of the stimuli or the escalation of inflammation [55]. One of these mechanisms of IL-1 β release involves the protection of interleukin within microvesicles shed from the plasma membrane [56] or secreted exosomes [57]. However, this early and rapid mechanism of IL-1β protected release is followed by a slower but non-protective IL-1β liberation from the cells [55]. On the basis of this information, we speculate that the contact of THP-1 cells with a mixed biofilm triggered several interleukin release mechanisms, allowing its protection from rapid degradation by the gingipains.

With regard to TNF-α, its production was triggered by THP-1 mainly in response to *P. gingivalis* treatment but contact of these with a mixed biofilm eliminated this phenomenon. The explanation could be again found in the proteolytic properties of the gingipains, as TNF-α is the substrate for RgpA, RgpB, and Kgp but with different proteolytic kinetics [58]. Our present results suggest that the activation of proteolysis is specifically performed by RgpA. The increased expression of the genes encoding RgpA, RgpB, and HagA supported this observation. The protective or stimulating properties of fungal biofilm we observed and presented above, can influence the viability and thereby the proteolytic efficiency of bacterial cells. Moreover, in our previous study, we found that the composition of biofilm formed by those two species was stabilized by specific binding among the microbial surface proteins, in particular between RgpA and Als3 [34]. This close interaction between the hemagglutinin domain of Rgp and Als3, exposed on the surface of fungal hyphae, may perhaps influence the activity of the gingipain catalytic domain, resulting in the efficient digestion of the cytokine.

The critical role of gingipains in the regulation of interleukin expression by THP-1 upon contact with mixed-species biofilm was further confirmed by the IL-8 secondary cytokine responses. Supernatants from either pathogen in contact with THP-1 cells as a mono-species biofilm triggered the intensive production of IL-8, whereas the mixed-species biofilm in the same situation caused an almost complete disappearance of this cytokine. It is well known that IL-8 is very prone to gingipain digestion, performed mainly by RgpA [59]. Our identified rapid degradation of IL-8 could be, again, explained by the increased presence or activity of gingipains in the environment, with a consequential modulating effect on the host immune response and the lower influx of host defense cells to the place of infection [47,60].

The reduced response of macrophages to contact with mixed biofilm thus may be a consequence of the strict collaboration of these microorganisms for self-protection. The level of this cooperation is variable and can depend on the progress of biofilm development. This process undoubtedly also requires precise regulation, as evidenced by a previously observed decline in fungal cell viability with an excessive increase in *P. gingivalis* proteolytic activity and bacterial cell count [34].

The mixed-species biofilm formed between *P. gingivalis* and *C. albicans* during the course of periodontal disease is part of the repertoire of known cooperative interactions previously identified for fungi and other bacteria species such as *Streptococcus oralis*, *S. sanguinis*, or *S. gordonii*. These collaborative interactions have also been shown to augment tissue invasion and colonization compared with monotypic infections [61].

Our current in vivo murine model study supported our initial in vitro findings that the fungal biofilm has a protective function for invading bacteria. Moreover, our observations suggested that a somewhat local infection was promoted in this way at the biofilm formation state since bacterial infiltration of other organs was limited. The local survival and activity of the bacteria increased under the condition of mutual contact between both pathogens, while the number of fungal cells also increased significantly. This suggested that stimulatory processes or cross-responses between microorganisms exist and that the gingipains are likely involved. Such conclusions can be drawn from our in vitro observations concerning the increase in mixed biofilm development, where the fungal cells dominated numerically but the biofilm showed an increase of gingipain activity. These results indicated that under adverse conditions, the bacteria were protected from death and that fungal presence could also enable the initiation of further infection, especially if the host defense system had a reduced efficiency. Furthermore, the ability of bacteria to degrade major defense molecules through gingipain action may as a consequence lead to an intense local infection. However, much more information can be obtained in future studies by conducting these analyses over a longer time scale.

The question arises as to whether the bacterial cells are hidden within the biofilm structure during the formation of a mixed biofilm, and thus protected by matrix components from host recognition or if mixed biofilm components are involved in diminishing bacterial recognition signals and host defense cell responses. The role of the gingipains seems to be particularly interesting as these enzymes can not only modulate the host responses but also can determine the structure of a mixed biofilm.

A poly-species infection can occur in different ways involving the simultaneous or sequential action of pathogens on host cells. Using gingival fibroblasts isolated from HD and from periodontic patients (PPs) in our current study, we considered the consequence of subsequent fungal infection and the possible impacts on the host responses.

Similar to immune cells, fibroblasts use specific sensory receptors to recognize different microorganisms and their surface-exposed molecules (pathogen-associated molecular patterns, PAMPs). Among these factors, the evidence to date has revealed the constitutive expression of selected Toll-like receptors such as TLR2, TLR3, and TLR4 as well as the adhesion molecules ICAM-1 (CD54) and CD44 [62] which are responsible for *C. albicans* cell recognition [63,64]. The differences in fungal virulence factor sensing by HD and PP fibroblasts, as reflected by the varying cytokine production by both types of host cells, resulted in a significantly lower response of the PP fibroblasts. The explanation for this finding may be that the prolonged contact with highly destructive *P. gingivalis* proteases resulted in chronic changes to the composition and stability of the host cell surface and receptors that are important for fungal cell recognition. It was suggested by Wilensky et al. [65] that gingipains selectively reduce the surface expression of the innate immune receptor CD14 and thereby altered the responsiveness of host cells. Interestingly, the reduction of CD14 expression was found in that same study to depend on the hemagglutinin/adhesion properties of gingipains and cause a hypo-responsiveness to bacterial challenge [65]. However, further studies are needed to clarify the mechanism of retaining of such changes by host cells [66,67].

In summary, we conclude that the co-infection of *P. gingivalis* with *C. albicans* can lead to a milder inflammation at the place of infection. Gingipains appear to be particularly involved in this process, as their increasing activity has been reported in mixed-species biofilm. The question, whether this is the result of regulation of the enzyme activities or an increase in bacterial survival, remains still open.

In an in vivo model, the protection of bacterial cells by *C. albicans* enabled these bacteria to persist for a longer time in the chamber fluid which may indicate a chronic nature of the co-infection process. This will be crucial for host vitality, especially if there is a weakened immunity that can facilitate infection renewal. On the other hand, a mixed biofilm will be more resistant to antibiotic treatment, also favoring the spread of infection.

## 4. Materials and Methods

### 4.1. Ethical Approval

Human studies were approved by and carried out in accordance with the recommendations of the Bioethical Committee of the Jagiellonian University in Kraków, Poland (permit numbers 1072.6120.12.2017, Approval Date:18 May 2017). All animal studies were performed in accordance with European Union regulations for the handling and use of laboratory animals and the protocols were approved by our Institutional Animal Care and Use Committees (Jagiellonian University, Krakow, Poland; Permit number 43/2018).

### 4.2. Host Cell Cultures

#### 4.2.1. Macrophages

The human acute monocytic leukemia cell line THP-1 was cultured in RPMI 1640 medium supplemented with 10% heat-inactivated fetal bovine serum (FBS), 2 mM l-glutamine, 100 U mL^−1^ penicillin, and 100 mg mL^−1^ streptomycin (all compounds purchased from Cytogen, Zgierz, Poland) at 37 °C in a humidified atmosphere of 95% air and 5% CO_2_. To facilitate their differentiation to macrophage-like cells, THP-1 cells were seeded into 12 well tissue culture plates at a density of 10^6^ cells/well in complete medium and treated with phorbol 12-myristate 13-acetate (PMA; Sigma–Aldrich, Saint Louis, MO, USA) at a final concentration of 10 ng mL^−1^ for 48 h. The progress of differentiation was monitored using a Scepter Handheld Automated Cell Counter (Merck Millipore, Burlington, MA, USA). After cell differentiation, the fresh culture medium without antibiotics was added with the preservation of PMA content for the entire duration of experiments. Inoculation with selected microorganisms was performed after 24 h of cell incubation under these conditions.

#### 4.2.2. Primary Fibroblasts

Fibroblasts were isolated from gingival tissue specimens as described previously [68]. Briefly, the cells were collected from healthy individuals undergoing orthodontic treatment (*n* = 6) and from patients with chronic periodontitis (*n* = 5) at our facility. The connective tissue remaining after enzymatic separation of the epithelial layer was digested overnight with 0.1% collagenase I (Invitrogen, Carlsbad, CA, USA) at 37 °C. After extensive washing with PBS, the cells were cultivated in Dulbecco’s Modified Eagle Medium (DMEM; PAA Laboratories, Thermo Fisher Scientific, Waltham, MA, USA) supplemented with 10% heat-inactivated FBS (PAA Laboratories, Thermo Fisher Scientific, Waltham, MA, USA), penicillin/streptomycin (50 U mL^−1^), gentamicin (50 U mL^−1^), and until passage 2 with nystatin (10 μg mL^−1^, PAA Laboratories, Thermo Fisher Scientific, Waltham, MA, USA), at 37 °C in a humidified atmosphere containing 5% CO_2_. At one day prior to and during the experiments, fibroblasts were cultured in antibiotic and antimycotic free DMEM containing 2% FBS. The homogeneity and viability of the fibroblast cultures were determined according to a previously described standardized procedure [69] following staining of the cells at their second passage with anti-vimentin and anti-cytokeratin antibodies. Cell viability was confirmed to be 95%–98% by trypan blue staining. The fibroblast cultures at passages 3–9 were used in the subsequent experiments.

### 4.3. Microbial Cultures

A wild-type strain of *P. gingivalis*, W83 (was obtained from Prof. Jan Potempa, Jagiellonian University in Krakow) was grown in anaerobic cultures for 5–7 days at 37 °C on blood agar plates as described previously [68]. The bacteria were then inoculated into brain–heart infusion (BHI) broth (BD Biosciences, Franklin Lakes, NJ, USA) supplemented with 0.5 mg mL^−1^ cysteine, 10 µg mL^−1^ hemin, and 0.5 µg mL^−1^ vitamin K and cultured overnight in an anaerobic chamber (85% N_2_, 10% CO_2_, and 5% H_2_). After washing with phosphate buffered saline (PBS), the bacteria were counted, and a new culture was started in fresh BHI broth for 20 h. The initial cell count was determined by OD measurement at 600 nm, performed for the diluted liquid culture and assuming that an OD_600_ value of 1 corresponded to 10^9^ colony-forming units (CFU) per mL.

For the isolation of cell wall components, *C. albicans* (ATCC10231) hyphae were obtained by cell propagation at the same conditions for 72 h with constant shaking.

### 4.4. Preparation of C. albicans Cell Wall Components Including Mannans and Glucans and Cell Wall Proteins

The isolation of fungal cell surface compounds (i.e., mannans and glucans) was performed in accordance with previously described methods [70]. The mannans were precipitated with Fehling’s reagent from the supernatant acquired during autoclaving the fungal cells at 121 °C in 20 mM citrate buffer. Glucans were isolated by heating the remaining cell pellet in the isopropyl alcohol (80%) under reflux. Any proteinaceous contamination of the polysaccharide preparations was removed by treatment with 250 μg of proteinase K (Sigma–Aldrich, Saint Louis, MO, USA) for 4 h at 55 °C in 20 mM Tris-HCl. The concentration of mannans or glucans was determined using the phenol–sulfuric acid method [71,72], modified for 96 well microtiter plates [73]. About 200 ng of glucans and 80 ng of mannans could be obtained from 10^5^ fungal cells.

The cell wall proteins were isolated from the hyphal form of *C. albicans* [73] by cell treatment with β-1,3-glucanase (Quantazyme; Qbiogene, Carlsbad, CA, USA) in the presence of 40 mM 2-mercaptoethanol. The supernatant obtained after cell removal containing released cell surface proteins was further purified by ion chromatography on MonoQ-Sepharose (GE Healthcare/Pharmacia, Uppsala, Sweden) to eliminate DNA contamination.

### 4.5. Recombinant Sap Production

Selected Sap isoenzymes were obtained according to the method described previously by us and others [74,75] after their overexpression in the *Pichia pastoris* system (Invitrogen, Carlsbad, CA, USA). The purity of the enzymes was determined by sodium dodecyl sulfate-polyacrylamide gel electrophoresis (SDS-PAGE), and their proteolytic activities were assayed using fluorescent casein, stained with fluorescein derivate (Invitrogen, Carlsbad, CA, USA) in 0.1 M buffers at pH values that were appropriate for each enzyme [74].

### 4.6. Mixed-Species Biofilm Formation for Gingipain Detection

Two methods of mixed biofilm formation were utilized, depending on the subsequent analysis. For the testing of gingipain production, *P. gingivalis* cells (CFU:10^8^ cells mL^−1^) and increasing amounts of *C. albicans* cells (CFU = 10^6^, 10^7^, 10^8^ cells mL^−1^) were cultured simultaneously in RPMI supplemented with 10% FBS and 10 ng PMA for 24 h in a humidified atmosphere of 95% air and 5% CO_2_ (normoxia), or in a GENbox jar anaerobic generator (anoxia) (bioMérieux, Craponne, France). As references, single-species cell propagation was used. After cell removal, the supernatant was collected, and immediately the gingipain activity was determined.

To analyze the expression of genes that encode selected gingipains (i.e., the arginine-specific genes *rgpA*, *rgpB*, the lysine-specific gene *kgp*, and the hemagglutinin component of the gingipains *hagA*), the microbial cells were cultured under the same conditions for 3 h at MOI 1:1 (10^8^ cells mL^−1^). The cell pellets were used for mRNA extraction.

### 4.7. Determination of Gingipain Activity

The proteolytic activities of gingipain R (Rgp) in the biofilm supernatants were determined by monitoring the hydrolysis of the chromogenic substrates benzoyl-l-arginine-*p*-nitroanilide (BApNA), as previously described [76]. Briefly, 130 μL of assay buffer (200 mM Tris-HCl, 100 mM NaCl, 5 mM CaCl_2_, pH 7.6) supplemented with fresh 10 mM l-cysteine was mixed with 45 μL of supernatant. The mixtures were then incubated at 37 °C for 15 min prior to the addition of 25 μL of 4 mM substrate in assay buffer. The formation of *p*-nitroanilide was measured as an increase in optical density at 405 nm over a 120 min period using the Power Wave X Select microplate reader (BioTek Instruments, Winooski, VT, USA).

### 4.8. Western Blot Analysis

For immunoblotting analysis, the proteins from the supernatants obtained after centrifugation were separated by SDS-PAGE and transferred to a PVDF membrane (Merck Millipore, Burlington, MA, USA) for 90 min at 250 mA in a 2.5 mM Tris, 19.2 mM glycine buffer with 20% methanol. The membrane was then incubated with primary monoclonal rabbit antibodies against Rgp A (20 μg/mL) followed by anti-rabbit horseradish peroxidase-labeled secondary antibodies (Cell Signaling Technology, Danvers, MA, USA). Protein bands were visualized with luminol-based chemiluminescence substrate (Cyanagen, Bologna, Italy). The signals were captured and quantified using a ChemiDoc XRS+ (Biorad, Hercules, CA, USA).

### 4.9. RNA Isolation, Reverse Transcription, RT-PCR, and RT-qPCR

Total RNA from *P. ginvivalis* strains was extracted using the method of Chomczynski and Sacchi [77] with a DNase I digestion step. The RNA quantity was measured using a NanoDrop spectrophotometer (Thermo Fisher Scientific, Waltham, MA, USA). The cDNA was synthesized using 1 μg of total RNA, M-MLV reverse transcriptase (Promega, Madison, WI, USA), and oligo(dT)_15_ primer in accordance with the manufacturer’s recommendations. The RT-qPCR was then performed on an Eco Real-Time PCR System (Illumina, San Diego, CA, USA) using a SYBR qRT-PCR Kit (A&A Biotechnology, Gdynia, Poland) in a 10 μL reaction volume under the following cycling conditions: an initial denaturation at 95 °C for 5 min and 40 cycles of denaturation at 95 °C for 20 s, primer annealing at 56 °C for 15 s, and extension at 72 °C for 20 s. Each sample was analyzed in duplicate and the expression levels were normalized to those of 16S rRNA mRNA. Primers for RT-qPCR were obtained from Genomed (Warszawa, Poland) and are listed in Table 1. Relative gene expression was analyzed by the ΔΔCq method [78].

### 4.10. Host Cell Infections

#### 4.10.1. THP-1 Co-Culture with Two Microbial Species

For the study of host responses to single- and mixed-species biofilms, THP-1 macrophage-like cells were incubated with *P. gingivalis* and *C. albicans* cells or supernatants collected from formerly formed biofilms, under the aforementioned conditions at a multiplicity of infection (THP-1:yeast:bacteria ratio) of 1:1:100 for 3 or 24 h, at 37 °C. After cell removal by centrifugation (200× *g*, 5 min), the supernatants were used to assay the cytokine production by the host cells.

#### 4.10.2. Fibroblast Challenge with Virulence Factors of *C. albicans* Cells

Fibroblasts were cultured at a density of 0.5 × 10^5^ cells/well in a six-well culture plate and maintained in 2 mL of DMEM supplemented with 10% FBS. After reaching confluence and washing twice with fresh medium, the fibroblasts were treated for 6 h at 37 °C in antibiotic-free medium with *C. albicans* purified virulence factors: glucans and mannans (1 µg/mL), cell wall proteins (100 ng/mL), and selected aspartic proteases (0.2 µg/mL). The determination of cytokines in the collected cell supernatants was performed similarly to THP-1 cytokine analysis.

### 4.11. Cytokine Assay

The supernatants collected after a 3 and 24 h incubation of macrophage-like THP-1 cells with a mono- or dual-species biofilm containing *P. gingivalis* and/or *C. albicans* cells at the multiplicities specified above were used for the determination of select interleukin concentrations. IL1β, TNF-α, and IL-8 were analyzed by enzyme-linked immunosorbent assay (ELISA) using a commercially available kit (BD OptEIA; BD Biosciences). The absorbance values were read at 450 nm.

### 4.12. Subcutaneous Chamber Model

A subcutaneous chamber model was employed to examine the response of the host organism to mono- or dual-species infections [48,49]. Experiments were performed on female C57BL/6 (6−8 weeks old; 22−25 g) mice obtained from Jackson ImmunoResearch Laboratories (Cambridge, United Kingdom). Briefly, a surgical-grade titanium wire coil (diameter 5 mm) was implanted subcutaneously into the dorsal-lumbar region of each animal. After a healing period of 10 days, *P. gingivalis* cells (10^8^ CFU/mL), *C. albicans* cells (10^6^ CFU/mL), or a mixture of both types of microbial cells were injected into the chambers, with PBS used as a control reference. For the analysis of sequential infections, the *C. albicans* cells were injected at one day prior to the bacterial cell injection. All living animals (two sets, six animals per set, with two additional animals, treated with PBS and served as a control) were used to calculate the % of survival. The chamber fluids were aspirated at different time points (24–96 h post-inoculation) and used to calculate the recovered CFU and neutrophil enzyme activity. Estimation of the bacterial load in the chamber and distribution to the organs was performed twice, in two independent experiments (6 animals for each repetition, 6−8 weeks old; 22−25 g). The chamber fluids were collected only from live animals at different time points (24–96 h post-inoculation). Twenty microliters of fluid were aspirated 24, 48, and 72 h post-infection. After four days (96 h p.i.), the survived animals were euthanized and all fluid from the chamber was collected. Ten microliters of chamber fluid were used for enumeration of recovered bacteria and/or fungi, 10 μL for analysis of the activity of neutrophils’ enzymes (5 μL for NE activity, 5 μL for MPO activity). After euthanization, the blood was collected to assay the IL-6, IL-10, MCP-1, IFN-γ, and TNF-α levels in plasma using a cytometric bead array mouse inflammation kit (BD Biosciences). Blood analysis was performed using the Scil Animal Care Company ABC Hematology Analyzer. The spleen and kidneys were homogenized to estimate the microorganism distribution. Homogenates and chamber fluids were plated onto blood agar plates for *P. gingivalis* analysis or YPD plates grown in aerobic conditions at 30 °C for *C. albicans.*

### 4.13. Determination of Elastase and Myeloperoxidase Activities

Neutrophil influx into the chamber was analyzed using MPO and NE activity measurement as markers. MPO activity (chlorination and peroxidation processes) was measured using an EnzChek Myeloperoxidase Activity Assay Kit (Invitrogen, Carlsbad, CA, USA). Briefly, to assess chlorination activity, 50 µL standards of MPO and 1:20 diluted chamber fluids were mixed with 50 µL 2× 3′-(*p*-aminophenyl) fluorescein (APF) working solution and incubated for 30 min in darkness at room temperature. The reactions were stopped by adding 10 µL of 10× chlorination inhibitor to all samples and the standards, and changes in fluorescence were measured at 485 nm excitation and 530 nm emission. For peroxidation activity, 50 µL of MPO standards and 1:20 diluted chamber fluids in PBS were mixed with 50 µL of 2× Amplex^®^ UltraRed reagent working solution and incubated at room temperature for 30 min, protected from light. The reactions were stopped by adding 10 µL of 10× peroxidation inhibitor to all samples, and any increase in fluorescence was detected using an excitation wavelength of 530 nm and emission at 590 nm.

The NE activity was analyzed using an EnzChek Elastase Assay Kit (Invitrogen, Carlsbad, CA, USA). Fifty microliters of reaction buffer (0.1 M Tris-HCl, pH 8.0) added to 50 µL of DQ elastin substrate and then mixed with 100 µL of each sample containing chamber fluids diluted 1:40 in reaction buffer-). Porcine pancreatic elastase (0.25 U/mL) was used as a positive control. After 30 min of incubation in darkness at room temperature, fluorescence increases were measured at 505 nm excitation and 515 nm emission.

### 4.14. Statistical Analysis

All experiments were performed at least in triplicate, and the results are expressed as the mean ± SEM. Statistical comparisons were performed using Prism 5.0 software (GraphPad, San Diego, CA, USA) using two-tailed Student *t*-tests or one- or two-way factorial analyses of variance (ANOVA) followed by Bonferroni post-tests. Differences were considered significant at *p* < 0.05.

## Figures and Tables

**Figure 1 ijms-21-01984-f001:**
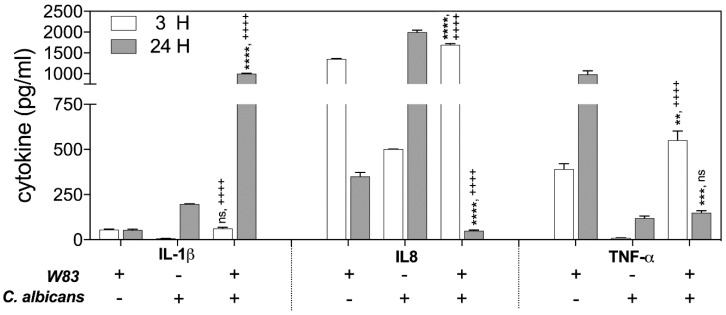
Changes in the protein production of the indicated cytokines in monocyte-like THP-1 cells under contact with supernatants acquired from biofilms formed by fungal (*C. albicans*) and bacterial (*P. gingivalis*) pathogens. THP-1 monocyte-derived macrophages were incubated with supernatants obtained from a mixed-species biofilm formed for 3 and 24 h by *P. gingivalis* and *C. albicans* cells (MOI = 100:1) or from a single-species biofilm of these microbes. The protein production level was analyzed by ELISA. The signal from appropriate references, where the THP-1 cells were treated with the fresh medium, was subtracted from the values measured for cells treated with supernatants. The experiment was performed in triplicate in three independent repetitions. The data from a representative repetition are expressed as the mean ± SEM. One-way ANOVA with a Bonferroni’s multiple comparisons test was used to determine the statistical significance levels for comparisons of mixed biofilm versus bacteria or fungi and marked with (*) or (+), respectively. Number of asterisks or crosses denote statistical significance (*p* > 0.1234 ns, ** *p* ≤ 0.0021, *** *p* ≤ 0.0002, **** *p* < 0.0001).

**Figure 2 ijms-21-01984-f002:**
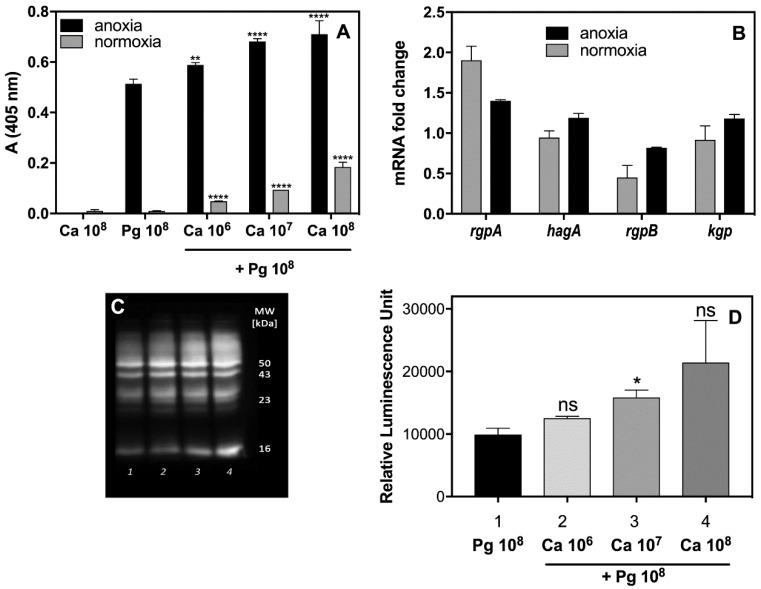
Changes in the gingipain activity and protease production by *P. gingivalis* growing within a *C. albicans* biofilm. (**A**) In assays for Rgp activity and (**C**,**D**) protein expression, the *P. gingivalis* cells grew (1) alone or in the presence of fungal cells at an MOI of (2) 100:1, (3) 10:1, (4) 1:1 for 24 h. The activity of Rgp was determined using BAPNA as the substrate followed by absorptiometric monitoring of enzyme kinetics. (**B**) qRT-PCR analysis of gingipain-encoding genes was performed for bacterial cells that formed a typical biofilm with *C. albicans* cells at an MOI of 100:1 during 3 h of mutual contact between both microorganisms. (**C**) Protein expression was analyzed by Western blotting and chemiluminescence detection with (**D**) respective quantification of the signals by phosphoimager. Each experiment was performed three times in triplicate, with the presentation of the representative data set. The gene expression was analyzed with one-way ANOVA with Sidak’s multiple comparisons test. Asterisks denote statistical significance (*p* > 0.1234 ns, * *p* ≤ 0.0332, ** *p* ≤ 0.0021, **** *p* < 0.0001).

**Figure 3 ijms-21-01984-f003:**
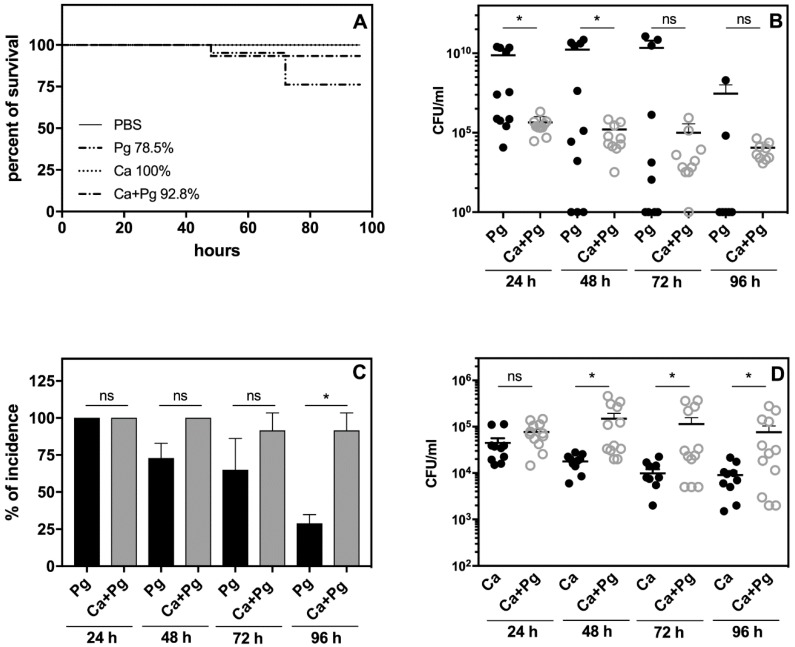
Changes in (**A**) mouse survival, (**B**) bacterial or (**D**) fungal viability, and (**C**) bacterial infection incidence in the subcutaneous chamber fluid. Mortality is expressed as a percentage and was zero in the control group. The graph depicts the mortality outcomes in the mice after bacterial (10^7^ CFU in 0.1 mL PBS), fungal (10^5^ CFU in 0.1 mL PBS) or mixed microbial exposure using a subcutaneous chamber model of infection. Subcutaneous chambers were initially inoculated with *P. gingivalis*, *C. albicans* or a mixture of these strains as described in the methods section. For control animals, we applied inoculation of the chambers with PBS. Recovery of viable bacteria or fungi from these chambers (CFU/mL) was determined by colony counting on agar plates following propagation at the conditions appropriate for the respective microorganism. Each dot represents a single animal. The data were collected from 2 sets of experiments. A Mann–Whitney U test (*n* = 8 mice for each group, with 2 as references) was performed for statistical analysis. Bars represent the mean CFU/mL of 12 chambers and include a standard error of the mean. Asterisks denote statistical significance (*p* > 0.1234 ns, * *p* ≤ 0.0332).

**Figure 4 ijms-21-01984-f004:**
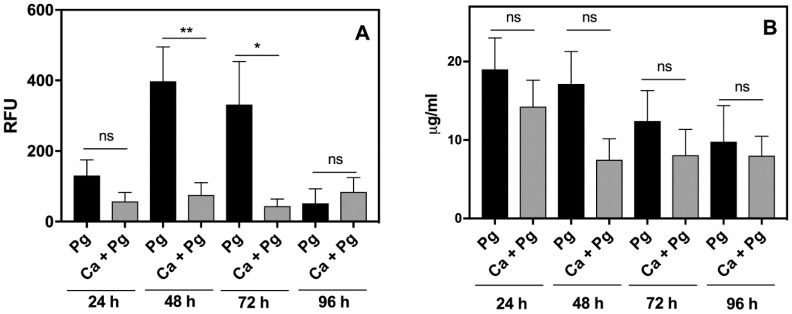
Activities of (**A**) the neutrophil enzymes elastase and (**B**) myeloperoxidase in the chamber fluid. Neutrophil elastase (NE) activity was measured in chamber exudates at selected time points using an EnzChek elastase assay kit. The appearance of the product was detected fluorometrically. Myeloperoxidase (MPO) activity was determined using its peroxidase properties and EnzChek MPO activity assay kit. Data are representative of two independent experiments and are expressed as means ± SEM. Significance was calculated using an unpaired *t*-test. Asterisks denote statistical significance (*p* > 0.1234 ns, * *p* ≤ 0.0332, ** *p* ≤ 0.0021).

**Figure 5 ijms-21-01984-f005:**
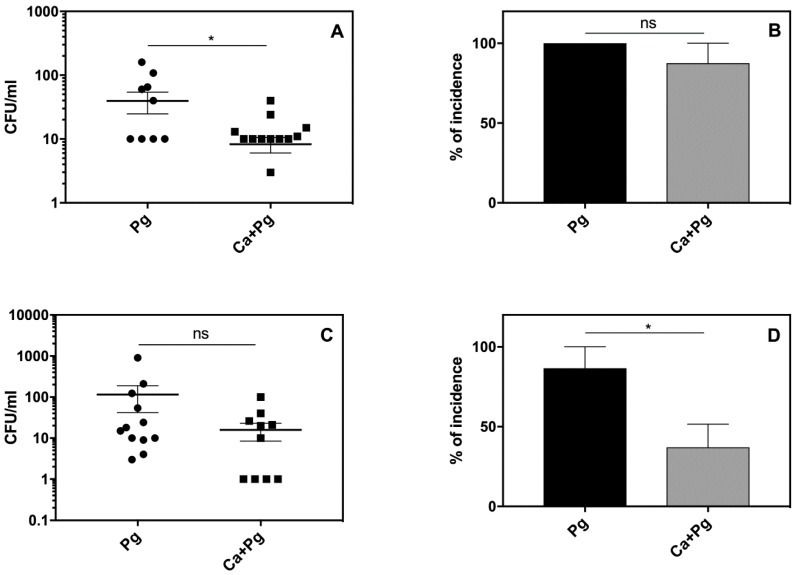
Infiltration of mouse organs by *P. gingivalis* at 96 h post-infection: (**A**,**B**) kidney; (**C**,**D**) spleen. After 4 days of the microbial challenge, the infection of isolated organs was analyzed by counting microbial colonies grown on appropriate agar plates. Two independent experiments were performed (*n* = 8 mice). Data for each tissue are expressed as a mean ± SEM. Asterisks denote statistical significance (*p* > 0.1234 ns, * *p* ≤ 0.0332).

**Figure 6 ijms-21-01984-f006:**
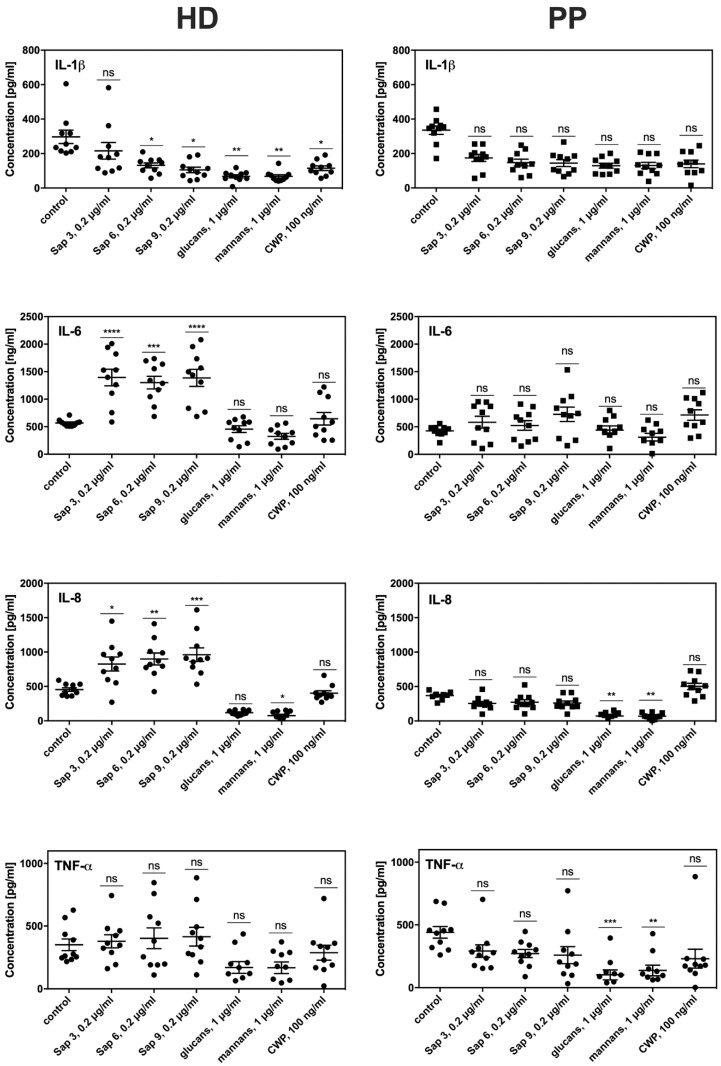
Cytokine production by fibroblasts from (**HD**) healthy donors and from (**PP**) periodontitis patients (right panels). The secretion of specific cytokines was identified by ELISA in the growth medium of fibroblasts that contacted with different fungal virulence factors for 6 h. The experiments were performed in duplicate in three independent sets. Data from a representative set were presented. Asterisks denote statistical significance (*p* > 0.1234 ns, * *p* ≤ 0.0332, ** *p* ≤ 0.0021, *** *p* ≤ 0.0002, **** *p* < 0.0001).

**Table 1 ijms-21-01984-t001:** qRT-PCR primers used in this study for the identification of gingipain gene expression.

Gene	Primer Sequence
16S rRNA For	TGTAGATGACTGATGGTGAAAACC
16S rRNA Rev	ACGTCATCCCCACCTTCCTC
*rgpA* For	TGGACAGGTTGTAAACTTTGCGCC
*rgpA* Rev	TTGCCTTGTTCCGAAGTTTCGCTC
*kgp* For	GCTCAGTACATCCTGCAGAAGTTC
*kgp* Rev	CTATAAGAAGCCTGATTCTGAGGC
*hagA* For	TGATGACGTGGCTGTTTCTGGTGA
*hagA* Rev	TTGTACTGGCCGGGAGCTACATTT
*rgpB* For	AAGATATCTATAAGAGCGTCTTCA
*rgpB* Rev	CGACCGATGAAGACTTCG

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
