# Peer review of "Candida albicans Shields the Periodontal Killer Porphyromonas gingivalis from Recognition by the Host Immune System and Supports the Bacterial Infection of Gingival Tissue"

_ijms, 2020, doi:10.3390/ijms21061984_

Round 1

Reviewer 1 Report

The manuscript focuses on the role of fungal-bacterial interactions in periodontal disease. Specifically, the authors explore the effect of the interaction between Porphyromonas gingivalis, a main causative agent of periodontitis, and Candida albicans, the most common fungus isolated from the oral cavity, on the human host. The authors check carefully how the mixed species and some of their products (gingipain, cell wall components and enzymes) influence host responses and the outcome of infection. The manuscript features multiple complex and relatively well designed experiments and manage to answer some important questions about the nature of the explored interaction. As such the article presents valuable new information to the field. However, I do recommend to consider the following points that should improve manuscript quality and scientific value.

1.       Overall, the manuscript is difficult to read. The presented information is not very well constructed and topics are switched often. The length of the sentences also contributes to this issue. Aside from these, I also noted multiple styling mistakes (e.g. line 24 abstract).

2.       Although the list of performed experiments is very impressive, I am afraid that the reader is forced to understand every detail and at the end fails to see the overall message from this work (perhaps also related to my point 1.). Mainly, it remains unclear to me whether the overall protective effect of Candida presence on P. gingivalis virulence is due to modulating  pingipain production in the host/infection models, due to physical separation from the host cells or both.

3.       It was unclear to me whether the experiment in Figure 1 was done with pre-grown biofilm cells, planktonic cell co-culture or supernatant. This information can make a big difference in data interpretation.

4.       Regarding Figure 3 I am not certain that CFU for C. albicans is a suitable approach for estimation of cell numbers. Host-derived signals stimulate hyphal growth in the fungus. Hyphal filaments are impossible to separate by common techniques (vigorous vortexing or even sonication) and 1 CFU typically represents more than 1 cell. Another issue with this figure is panel A. Here it is not clear how the % survival was calculated as the listed percentages cannot be achieved with sample size of 12. This was not described in materials and methods and only listed in the figure legend. Thus, it mandatory that the authors provide all the information required to follow the presented data or replicate the experiment. Missing/insufficient information (e.g. Fig 2 C) to interpret the figures and missing control conditions (e.g. PBS controls in the mouse experiments) were found throughout the manuscript.

5.       Considering Figure 3 it is important to specify how were the chamber fluids analyzed and how many were used for each time point? In line 634 it is mentioned that mice were euthanized after four days – what about already dead animals? Where these animals analyzed immediately and which parameters where analyzed? To make this animal model more transparent and comprehensible I recommend to include a supplementary table listing each single animal used in this study with treatment, body weight, bacterial and fungal load, age, experimental day of death, etc.

6.       Finally, for Figure 3B there is first a big difference from Pg to Ca+Pg meaning a decrease of bacterial cells within the chambers. But, as C. albicans is claimed to protect P. gingivalis from host immune response this decrease is not clear to me as you also did not address this fact in your discussion. Second, from 48 to 96 h for Pg there are some zero values. I cannot imagine that in some of the chambers there is no single CFU of Pg.

7.       The data presented in Figure 6 is impossible to follow and interpret. This is also the only experiment where authors choose to use different wild type and a mutant C. albicans strain. Therefore, I am not sure how this fits to the manuscript and recommend to remove the data.

8.       In the discussion (page 13, line 421) you mention that gingipains are involved in stimulatory processes. You already showed this by the increased gingipain activity in mixed biofilms. Another option to confirm these results would be to repeat your experiments using gingipain deletion mutants.

All in all, I recommend that the authors streamline the information in the paper, include mandatory details about experimental design and data analysis, and discuss only relevant and key findings.

Author Response

Reviewer’s comments

REVIEWER 1:

1/ Overall, the manuscript is difficult to read. The presented information is not very well constructed and topics are switched often. The length of the sentences also contributes to this issue. Aside from these, I also noted multiple styling mistakes (e.g. line 24 abstract).

We have partially rewritten the text, noting the shortening of difficult to understand sentences. We marked all the changes within the text. We also subjected the manuscript to language correction at Boston BioEdit to remove stylistic and language errors. Making changes, we tried to improve the logical structure of the text.

2/ Although the list of performed experiments is very impressive, I am afraid that the reader is forced to  understand every detail and at the end fails to see the overall message from this work (perhaps also related to my point 1.).   is due to modulating  pingipain production in the host/infection models, due to physical separation from the host cells or both.

To make it easier to understand the text, we tried to simplify it as much as possible by eliminating unnecessary details (all changes marked in the text). We also emphasized the message of our work (highlighted in the new part of the discussion ) regarding the protective role of fungal biofilm for the functioning of bacterial cells.

However, it is difficult to attribute the protection of bacterial cells within mixed-species biofilm to a single factor. The increasing expression of rgpA that corresponds to an increase in protein concentration and gingipain activity is a good argument for it. The kgp expression did not change significantly but we also observed its increasing activity in the supernatants from biofilm (data not presented). Moreover, the host environment can also influence the formed biofilm, as we detected the increasing gingipain activity when the mixed biofilm was formed in FBS presence.  Hence, the protection of bacterial cells within the mixed-species biofilm does not necessarily concern only the modulation (increase) of gingipain production by the bacterial cells but also (or maybe in the dominant part) the increased potential to maintain the bacterial cell viability in unfavorable conditions — such evidence we presented in our previous work. And the increased  survival (depending on the amount of protecting fungal cells) can be manifested in the increased production of gingipains, at least in part. Of course, the possible physical separation from the host cells will also contribute to the sum of these effects. 

3/ It was unclear to me whether the experiment in Figure 1 was done with pre-grown biofilm cells, planktonic cell co-culture or supernatant. This information can make a big difference in data interpretation.

In our research, we carried out two combinations of these experiments. In the first, activated THP-1 cells remained in contact for 3 and 24 h with P. gingivalis and C. albicans cells. During this contact, the microbial cells formed a biofilm. The supernatants collected from THP-1 cells contacting biofilm were analyzed for the presence of cytokines, both at the genetic and at the protein level. However, the replications of these experiments generated a large spread of the results. Hence, we decided that in the next step, first, the stabile two-species biofilm would be formed, and then the soluble compounds, secreted during this process into the supernatant, would be used to stimulate THP-1 cells. However, such an approach significantly simplified the modeled situation. It eliminated not only the impact of human cells on the biofilm formation but also excluded part of the possible THP-1 responses that could result from the contact of THP-1 cells with the surface exposed virulence factors of microbial cells. This simplification allowed us to get more reproducible results. These results have been presented in the manuscript. Unfortunately, in their description, we did not maintain consistency and often used mistakenly the description suggesting the direct contact of THP-1 cells with planktonic or biofilm form of microorganisms. Thank you to the reviewer for bringing us to the attention of this mistake.

In the corrected text we already clearly refer to the usage of supernatants in the treatment of THP-1 cells.

4 A/ Regarding Figure 3 I am not certain that CFU for C. albicans is a suitable approach for estimation of cell numbers. Host-derived signals stimulate hyphal growth in the fungus. Hyphal filaments are impossible to separate by common techniques (vigorous vortexing or even sonication) and 1 CFU typically represents more than 1 cell.

We are aware that hyphal forms of C. albicans could be formed during incubation under the conditions examined, that is why we tried to select a good method for the analysis of cell numbers. The choice of the method was preceded by a comparison of three proceedings: (i) pipetting, routinely used in our laboratory, (ii) the method described by Bor et al. (Sci Rep. 2016 Jun 14;6:27956. Morphological and physiological changes induced by contact-dependent interaction between Candida albicans and Fusobacterium nucleatum. Bor B, Cen L, Agnello M, Shi W, He X), in which the sonication was applied prior to plating the cells on agar plates in order to disrupt aggregates, and (iii) the method described by Hayama et al. (Med Mycol. 2012 Nov;50(8):858-62. Cell preparation method with trypsin digestion for counting of colony-forming units in Candida albicans-infected mucosal tissues. Hayama K, Maruyama N, Abe S) based on the trypsinization of the aggregates. The results of colony counting are presented in the Figure below.  As shown, these results were quite comparable for the three tested methods. That is why we prepared our samples by pipetting with the greatest attention and precision, to properly disrupt the aggregates.

4 B/ Another issue with this figure is panel A. Here it is not clear how the % survival was calculated as the listed percentages cannot be achieved with sample size of 12. This was not described in materials and methods and only listed in the figure legend.

We fully agree with the reviewer as the calculation of % of survival was entered incorrectly. We analyzed obtained data once again, making the appropriate correction and adding data for a group of mice injected with PBS.

4C/Thus, it mandatory that the authors provide all the information required to follow the presented data or replicate the experiment. Missing/insufficient information (e.g. Fig 2 C) to interpret the figures and missing control conditions (e.g. PBS controls in the mouse experiments) were found throughout the manuscript.

We supplemented the necessary information about replicate in the legend to this figure and connect the data presented in Figure 2C and 2D (new Figure).

For mouse experiments, we used PBS injection as a control. For that purpose, we used a set of 8 animals. As the accumulation of serum in the chamber injected with PBS was limited due to the lack of local inflammation, thus we couldn’t obtain enough chamber fluid for CFU analysis, which would confirm the lack of P. gingivalis and/or C. albicans. This cohort of animals was used as a control to examine the survival of animals, bacterial distribution to the organs and the level of cytokines in serum. Relevant information was added in the Methods section

5A/ Considering Figure 3 it is important to specify how were the chamber fluids analyzed and how many were used for each time point? 5B/ In line 634 it is mentioned that mice were euthanized after four days – what about already dead animals? Where these animals analyzed immediately and which parameters where analyzed?

We clarified the procedures describing the animal model in the new version of the manuscript in the Methods section. However, we would like to respond directly to the Reviewer's question thus below we present the required information.

Estimation of the bacterial load in the chamber and distribution to the organs  was performed twice, in two independent experiments (6 animals for each repetition, 6−8 weeks old; 22−25 g). The chamber fluids were collected only from live animals at different time points (24–96 h post-inoculation). Twenty microliters of fluid were aspirated 24, 48 and 72 hours post-infection. After four days (96 h p.i.) survived animals were euthanized and all fluid from the chamber was collected. Ten microliters of chamber fluid were used for enumeration of recovered bacteria and/or fungi, 10 ul for analysis of the activity of neutrophils’ enzymes (5 ul for NE activity, 5 ul for MPO activity). Tissues and chamber fluids were not isolated from dead animals thus they were not analyzed. 

5C/ To make this animal model more transparent and comprehensible I recommend to include a supplementary table listing each single animal used in this study with treatment, body weight, bacterial and fungal load, age, experimental day of death, etc.

We agree with the reviewer; therefore, we clarified the description of the treatment of animals as listed above. Moreover, we also would like to present the reviewer the unpublished results, as animal body weight changes, blood cell morphology changes and the level of cytokines in the blood serum, to ensure the correctness of the performed tests.

6A/ Finally, for Figure 3B there is first a big difference from Pg to Ca+Pg meaning a decrease of bacterial cells within the chambers. But, as C. albicans is claimed to protect P. gingivalis from host immune response this decrease is not clear to me as you also did not address this fact in your discussion.

The results obtained using the chamber model should be considered comprehensively with the final consequence for the host and in  this respect, they confirmed previously published in vitro data showing P. gingivalis protection during biofilm formation with fungal cells (Karkowska-Kuleta et al., 2018, Bartnicka et al., 2019). Indeed, we observed a decrease of P. gingivalis cells in the presence of C. albicans early post-inoculation (within 48 h post-infection), when pathogens compete for the colonizing place. But it is followed by stabilization of P. gingivalis load in the chamber observed 72 and 96 h post-inoculation. Moreover, the analysis of P. gingivalis infected group of animals revealed a discrepancy in their response to bacteria. In some animals eradication pathogen (0 CFU) was observed already within 48 h, while in the presence of C. albicans we observed sustained presence of P. gingivalis up to 96 h post-infection.

B/ Second, from 48 to 96 h for Pg there are some zero values. I cannot imagine that in some of the chambers there is no single CFU of Pg.

As we used the same procedure of sample dilution for all chamber fluid samples, so, for some of the animals, we lose the information.

The data presented in Figure 6 is impossible to follow and interpret. This is also the only experiment where authors choose to use different wild type and a mutant C. albicans strain. Therefore, I am not sure how this fits to the manuscript and recommend to remove the data.

We did not decide to remove the data part, according to the reviewer's (1) suggestion, because the second reviewer had no objections to this data set.

Such observations have not been carried out so far. And it seems important to us to show that the fibroblasts of patients with periodontitis show considerable resistance to secondary infection and have a different sensitivity to the morphological form of fungus participating in this process.

In the discussion (page 13, line 421) you mention that gingipains are involved in stimulatory processes. You already showed this by the increased gingipain activity in mixed biofilms. Another option to confirm these results would be to repeat your experiments using gingipain deletion mutants

Yes, we performed such experiments, using treatment of host cells with extracts of biofilms formed between microbial cells, including gingipain depleted mutant strain (∆K∆RAB), used by us in previous experiments.  These results supported our findings regarding the significance of gingipains and their  role in the inactivation of selected interleukins (selected example of IL-8 production)

So, for wild type strain forming biofilm with fungus, we observed the summarizing effects:  interleukin production with the domination of its degradation.   On the other hand, for bacterial mutant strain (∆K∆RAB), we see the non-gingipain dependent stimulation of THP-1, with the high level of responses.

However, we did not decide to include these results in the manuscript to not complicated the pattern of interactions between host cells and formed biofilm.

All in all, I recommend that the authors streamline the information in the paper, include mandatory details about experimental design and data analysis, and discuss only relevant and key findings.

We have enriched the methodological information (highlighting the text in yellow) and shortened the discussion, as suggested by the reviewer.

Thank you to the reviewer for the constructive comments that allowed us to prepare a better version of our manuscript.

Reviewer 2 Report

This article is an extension of the authors’ work on effects of mixed biofilms of the oral bacterium P. gingivalis and the fungus C. albicans. This interaction between two microbes and a number of types of host cells is complex, so the results are a valuable contribution to our basic knowledge. Specifically, new information on interaction of C. albicans with fibroblasts from gingivitis and normal patients, and consequences of mixing the microbes in subcutaneous infection chambers are novel and potentially critical.  However, I have a serious reservation about the results shown in Figs. 1 and 2. Considering that three biological replicates were assayed in triplicate each, the error bars are significantly smaller than expected and customary for other labs for these types of experiments with multiple biological samples. Therefore, it is essential that primary data be presented to allow readers’ judgement about the probity of these results. Also, we need to see the values for un-exposed control THP-1 cells.

More minor comments by line number:

24 sentence fragment

28 insert “…including microbial exposure to a representative…”

31 and 32 delete “the” on both lines

48   delete “the” twice

55-56 That gingivitis causes these conditions appears to be an overstatement. The infection is associated with exacerbation of the diseases.

91-92. “However, the observation…” meaning not clear, please expand

122 “…dendritic cells.”

Fig. 1

--shows response to culture supernatants. This type of exposure greatly simplifies the analysis of the results, but the authors need to be careful to qualify that the response is thus to soluble microbial inducers

--values are missing for control cultures without stimulation

--comments on error bars given above

--W83 should be defined previously in the text

167 and 172 If these two paragraphs were switched in position, the order of the text would match the order in the figure, making it easier for the reader

172 and 177 The exposure has been to microbial supernatants, not to microbes

178-179 Meaning of the sentence is not clear, please expand

Fig. 2

2C give the origins of the samples in lanes 1-4

2D define RBI (y axis)

219 “…significantly reduced bacterial…” would be clearer

300 “efg1ΔΔcph1ΔΔ” is not standard nomenclature. Suggest efg1Δ cph1Δ/efg1Δ cph1Δ

Fig 6 Reading the figure will be easier if the columns are also labeled as HD and PP

Fig. 7 What does each data point represent? There are more data points than patients.

360 “…where supernatant-challenged…”

397-398 “Supernatants from either pathogen in contact with…”

405 This sentence seems overstated from the evidence. “…mixed biofilm thus may be a consequence…”

610 italicize C. albicans

766, 775 The citations for references 37 and 39 are incomplete

Round 2

Reviewer 1 Report

The manuscript focuses on the role of fungal-bacterial interactions in periodontal disease. Specifically, the authors explore the effect of the interaction between Porphyromonas gingivalis, a main causative agent of periodontitis, and Candida albicans, the most common fungus isolated from the oral cavity, on the human host. The authors check carefully how the mixed species and some of their products (gingipain, cell wall components and enzymes) influence host responses and the outcome of infection. The manuscript features multiple complex and relatively well designed experiments and manage to answer some important questions about the nature of the explored interaction. As such the article presents valuable new information to the field. However, I do recommend to consider the following points that should improve manuscript quality and scientific value.

  1. Overall, the manuscript is difficult to read. The presented information is not very well constructed and topics are switched often. The length of the sentences also contributes to this issue. Aside from these, I also noted multiple styling mistakes (e.g. line 24 abstract).
  2. Although the list of performed experiments is very impressive, I am afraid that the reader is forced to understand every detail and at the end fails to see the overall message from this work (perhaps also related to my point 1.). Mainly, it remains unclear to me whether the overall protective effect of Candida presence on P. gingivalis virulence is due to modulating pingipain production in the host/infection models, due to physical separation from the host cells or both.
  3. It was unclear to me whether the experiment in Figure 1 was done with pre-grown biofilm cells, planktonic cell co-culture or supernatant. This information can make a big difference in data interpretation.
  4. Regarding Figure 3 I am not certain that CFU for C. albicans is a suitable approach for estimation of cell numbers. Host-derived signals stimulate hyphal growth in the fungus. Hyphal filaments are impossible to separate by common techniques (vigorous vortexing or even sonication) and 1 CFU typically represents more than 1 cell. Another issue with this figure is panel A. Here it is not clear how the % survival was calculated as the listed percentages cannot be achieved with sample size of 12. This was not described in materials and methods and only listed in the figure legend. Thus, it mandatory that the authors provide all the information required to follow the presented data or replicate the experiment. Missing/insufficient information (e.g. Fig 2 C) to interpret the figures and missing control conditions (e.g. PBS controls in the mouse experiments) were found throughout the manuscript.
  5. Considering Figure 3 it is important to specify how were the chamber fluids analyzed and how many were used for each time point? In line 634 it is mentioned that mice were euthanized after four days – what about already dead animals? Where these animals analyzed immediately and which parameters where analyzed? To make this animal model more transparent and comprehensible I recommend to include a supplementary table listing each single animal used in this study with treatment, body weight, bacterial and fungal load, age, experimental day of death, etc.
  6. Finally, for Figure 3B there is first a big difference from Pg to Ca+Pg meaning a decrease of bacterial cells within the chambers. But, as albicans is claimed to protect P. gingivalis from host immune response this decrease is not clear to me as you also did not address this fact in your discussion. Second, from 48 to 96 h for Pg there are some zero values. I cannot imagine that in some of the chambers there is no single CFU of Pg.
  7. The data presented in Figure 6 is impossible to follow and interpret. This is also the only experiment where authors choose to use different wild type and a mutant C. albicans strain. Therefore, I am not sure how this fits to the manuscript and recommend to remove the data.
  8. In the discussion (page 13, line 421) you mention that gingipains are involved in stimulatory processes. You already showed this by the increased gingipain activity in mixed biofilms. Another option to confirm these results would be to repeat your experiments using gingipain deletion mutants.

All in all, I recommend that the authors streamline the information in the paper, include mandatory details about experimental design and data analysis, and discuss only relevant and key findings.

Author Response

Once again, we thank the reviewers for their time spent on helping us in preparing the improved version of the manuscript.

Reviewer 2 Report

The submitted changes and corrections are appropriate

Author Response

(The authors gave the same response as above.)
